# Beneficios mutuos de la enseñanza-aprendizaje máquina-humano

**Jorge Alvarado Díaz**
Universidad de Extremadura
jalvaradod@unex.es

**Elia Pacioni**
Universidad de Extremadura
HES-SO Valais-Wallis
elia.pacioni@hevs.ch

**Francisco Fernández De Vega**
Universidad de Extremadura
fcofdez@unex.es

## Abstract

Este trabajo muestra que una colaboración eficaz entre investigadores de Inteligencia Artificial (IA) y expertos y estudiantes de un dominio concreto -profesores de música y sus alumnos- produce beneficios mutuos a corto plazo y podrían inducir un círculo virtuoso de mejora a largo plazo: permite a los alumnos desarrollar sus capacidades de forma más eficaz y, a la vez esto influye en la mejora la calidad de los resultados de la IA. En concreto, mostramos cómo el número de ejercicios de armonía a 4 voces realizados por los alumnos aumentó notablemente, un 100 %, al utilizar una herramienta asistida por IA. En segundo lugar, las capacidades de la herramienta con IA para resolver ejercicios de armonía a 4 voces mejoraron considerablemente gracias al trabajo que profesores y alumnos habían desarrollado durante 4 años. Se analizaron más de 13.000 ejercicios de alumnos, lo que nos proporcionó ayuda para centrarnos en áreas más reducidas y prometedoras del espacio de búsqueda al aplicar algoritmos evolutivos (AE), lo que finalmente dio lugar a nuevas versiones del algoritmo capaces de encontrar soluciones sin errores en tiempos razonables.

**Keywords:** Mutación dirigida, Búsqueda local, Enseñanza automática evolutiva, Enseñanza humana, Armonización a 4 voces.

## 1. Introducción

En el paradigma de aprendizaje mediante la práctica, a diferencia del aprendizaje por observación, los estudiantes adquieren progresivamente experiencias útiles a medida que enfrentan e interactúan con los problemas abordados [1]. Este principio ha sido frecuentemente defendido en la forma de aprendizaje por ensayo y error. El término *práctica* a veces se entiende como la repetición de una tarea determinada. Por otro lado, el enfoque de aprendizaje sin errores se centra en la metodología de enseñanza. Como se describe en [2], enseñar a un alumno todos los prerrequisitos para una tarea inducirá un aprendizaje sin errores. Cuando ocurren errores, pueden corregirse identificando y enseñando los prerrequisitos faltantes. Así, el número de ejercicios que los estudiantes realizan –la repetición de la tarea– se correlacionará con la cantidad de errores que los profesores pueden detectar, permitiéndoles abordar los prerrequisitos necesarios. Por lo tanto, es crucial que los docentes involucren a los estudiantes con la materia de tal manera que estos estén dispuestos a realizar más ejercicios. Sin embargo, si se logra este objetivo, los profesores tendrán que dedicar más horas a revisar ejercicios, lo que puede convertirse en un problema en grupos grandes de estudiantes.

Consideramos aquí el caso de los estudiantes de música que, por primera vez, abordan la composición musical aprendiendo las reglas –prerrequisitos– de la armonía a cuatro voces: la construcción de las voces de Soprano, Alto, Tenor y Bajo a partir de una melodía dada. Los profesores deben primero explicar la gran cantidad de reglas que se deben aplicar –más de 50 reglas y excepciones–; en segundo lugar, cómo y cuándo aplicarlas; y finalmente, los estudiantes deben practicar dichas reglas

armonizando las melodías que el profesor les proporciona. Aunque se sigue el enfoque de aprendizaje sin errores, cuando los profesores explican las reglas necesarias para construir ejercicios de armonía, también se aplica el aprendizaje basado en la práctica. Dado el gran número de reglas y excepciones, la única manera de dominarlas es mediante la práctica, es decir, realizando la mayor cantidad posible de ejercicios. Esto plantea una hipótesis interesante: ¿podría la tecnología, como software de composición asistida por IA, ayudar a los estudiantes a recibir retroalimentación inmediata y reducir la carga de corrección para los profesores? Tal vez, el uso de herramientas interactivas podrían hacer que el proceso sea más dinámico, permitiendo a los estudiantes experimentar en tiempo real los efectos de sus decisiones armónicas.

Este artículo narra una historia de éxito que involucra a estudiantes de música, sus profesores e investigadores dedicados a la IA con un doble objetivo. Por un lado, buscamos mejorar la forma en que los estudiantes aprenden, alentándolos específicamente a realizar más ejercicios mediante el uso de una herramienta encargada de revisar automáticamente sus ejercicios de armonía a cuatro voces. De esta manera, los profesores ahorraron tiempo en la revisión de ejercicios y pudieron dedicar más tiempo a la enseñanza. Como mostramos a continuación, logramos aumentar en un 100 % la cantidad de ejercicios realizados por los estudiantes.

En segundo lugar, intentamos mejorar las habilidades de la herramienta Sharpmony [1] para la composición de música a cuatro voces. Esto fue posible al reducir el espacio de búsqueda gracias a 13.000 ejercicios recopilados en el proceso de enseñanza-aprendizaje desarrollado por profesores y estudiantes. Esta mejora permitió, por primera vez, resolver ejercicios complejos de armonía a cuatro voces. Además, todo el proceso que involucra el doble objetivo no se desarrolló en un enfoque de dos etapas. Por el contrario, se adoptó un enfoque iterativo y progresivo, en el que los ejercicios de los estudiantes ayudaban a mejorar la herramienta de IA, y estas mejoras, a su vez, permitían una mejor evaluación de los ejercicios de los estudiantes, lo que los motivaba a realizar más ejercicios, que luego se utilizaban para seguir mejorando la herramienta, y así sucesivamente. Este proceso sigue en funcionamiento dentro del proyecto Sharpmony.

El resto del documento está organizado de la siguiente manera: en la sección 2, describimos el problema abordado y presentamos varios enfoques previos. Luego, en la sección 3, explicamos la metodología. La sección 4 presenta los resultados y, finalmente, en la sección 5, exponemos nuestras conclusiones.

## 2. El problema de la armonización SATB y la forma en la que aprenden los alumnos

La armonización a cuatro voces, también conocida como SATB (Soprano, Alto, Tenor y Bajo), es una técnica ampliamente utilizada en la música coral. Popularizada por Bach en la era barroca, sus reglas evolucionaron y se perfeccionaron durante el clasicismo. Hoy en día, es una técnica que todo estudiante de música debe aprender. Los aspectos más importantes que se estudian en este tema son la progresión de acordes, la prohibición de ciertas combinaciones disonantes, los movimientos melódicos y el equilibrio sonoro entre las diferentes voces.

Los estudiantes matriculados en los conservatorios profesionales intentan resolver ejercicios de armonía que sus profesores les proporcionan de manera progresiva. Dada una melodía, que corresponde a la voz de soprano, los estudiantes deben completar las otras tres voces sin infringir ninguna de las reglas establecidas. El profesor se encarga, en primer lugar, de explicar las reglas que deben aplicarse; en segundo lugar, de proponer ejercicios; y, finalmente, de revisar esos ejercicios y proporcionar sugerencias para corregir los errores encontrados. La Figura 1 muestra un ejercicio revisado, en el que los errores detectados están marcados con colores. El ejercicio consta de cuatro voces que progresan como una serie de notas en sentido horizontal y, cuando se analizan verticalmente, corresponden a una serie de acordes, con cuatro notas por acorde, una por cada voz.

Aunque los profesores desearían que sus estudiantes realizaran más ejercicios, el problema en grupos numerosos es que el profesor necesitaría más tiempo, el cuál no siempre está disponible, para corregir todos los ejercicios realizados. Por lo tanto, desde un punto de vista pedagógico, sería de gran interés contar con herramientas que permitan a los profesores ahorrar el tiempo que dedican a la corrección de ejercicios, lo que les permitiría centrarse en el proceso de la enseñanza. Además, si los estudiantes

---

[1] https://sharpmony.unex.es

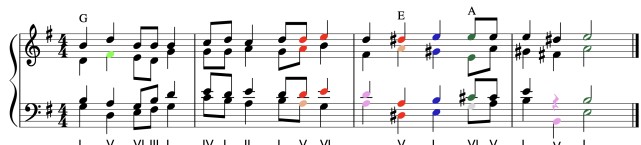

Figura 1: Ejercicio de armonía a cuatro voces con errores marcados. Por ejemplo, el color rojo indica quintas u octavas paralelas; el color oliva, un acorde incorrecto, etc.

pudieran realizar y corregir ejercicios de manera autónoma, no tendrían que esperar hasta la siguiente clase —generalmente la semana siguiente— para recibir la corrección de su profesor.

Uno de nuestros objetivos iniciales era el desarrollo de una herramienta de software, el proyecto Sharpmony [2], que incorporara enfoques evolutivos para resolver el problema de la armonización a cuatro voces. Sin embargo, nos dimos cuenta que la función encargada de evaluar la calidad de los resultados podría ser una herramienta útil para revisar los ejercicios de armonización a cuatro voces realizados por los estudiantes. Además, esto podría ayudar a los profesores a ahorrar tiempo en el proceso de revisión.

Analizamos a continuación cómo abordamos la armonización a cuatro voces desde el punto de vista de los AE y, posteriormente, su relación con el enfoque de enseñanza/aprendizaje en los conservatorios de música profesional.

## 2.1. Armonización a 4 voces evolutiva

Generar armonías a cuatro voces es un problema de interés desde el punto de vista de la IA y la creatividad computacional. De hecho, este fue nuestro punto de partida hace algunos años, cuando nos interesamos en abordar este problema mediante AE.

Los intentos de aplicar AE a la armonización a cuatro voces se remontan a la década de 1990, cuando se publicaron los primeros enfoques simplificados [3, 4]. Más recientemente, Kaliakatsos et al. intentaron generar una progresión de acordes asignando grados de escala a cada nota y luego seleccionando acordes y notas individuales de las voces en función de estos grados de escala [5].

En 2017, Fernández presentó un nuevo enfoque basado en algoritmos genéticos (AG) que consiguió evolucionar con éxito una partitura musical completa. Aplicando solo 11 reglas durante el proceso evolutivo, la partitura resultante tenía 10 errores, aunque se requirieron 24 horas para obtener este resultado [6].

De Prisco et al. presentaron EvoComposer [7], un AE para la armonización automática a cuatro voces. El sistema opera con un enfoque multiobjetivo, optimizando tanto la armonización mediante una selección adecuada de acordes como la calidad melódica de las líneas vocales.

En 2024, Pacioni and Fernández presentaron una nueva versión con más de 50 reglas y excepciones [8]. Aunque el aumento en el número de reglas acerca el sistema a la realidad práctica, también incrementa el tiempo computacional necesario para que el AG encuentre soluciones viables. Para abordar este problema, se añadió el operador de mutación dirigida para guiar el sistema hacia mejores soluciones, y se aplicó una evaluación anticipada y parcial de la función fitness para mejorar el rendimiento del algoritmo. También se introdujeron la mutación dirigida y la creación de modelos sintéticos [8], demostrando cómo la mutación dirigida puede contribuir a mejorar la convergencia del algoritmo. Sin embargo, hasta donde sabemos, el problema sigue abierto y, hasta la fecha, no se ha publicado una solución general del problema.

## 2.2. Aprendizaje-enseñanza humano-máquina

Si consideramos el punto de vista de los AE descrito anteriormente, la función fitness es la herramienta utilizada para corregir los ejercicios, ya que se encarga de evaluar la calidad de las soluciones candidatas. Hemos observado el interés de proporcionar esa herramienta específica, la función fitness, a los profesores con el fin de acelerar la corrección de ejercicios. Además, esto permitiría a los estudiantes revisar sus ejercicios sin la intervención del profesor tantas veces como quisieran, lo que

---

[2] https://sharpmony.unex.es/index.php?r=tutorials%2Fharmonic-manual

podría fomentar un mayor compromiso y, en última instancia, aumentar la cantidad de ejercicios realizados.

Por otro lado, si analizamos la forma en que los estudiantes aprenden, es evidente que invierten menos tiempo en resolver ejercicios con respecto a un AE. Mientras que los AE disponibles públicamente aplicados a la armonización a cuatro voces requieren varias horas para encontrar soluciones con un número reducido de errores, los estudiantes suelen necesitar menos de una hora para obtener un resultado similar. Y las preguntas que surgen son: ¿podría, de alguna manera, el proceso de aprendizaje de los estudiantes utilizarse para mejorar el enfoque evolutivo?

Esta pregunta fue propuesta por primera vez en [9], donde se presentó un enfoque relacionado con la Enseñanza/Aprendizaje Humano y la Enseñanza Evolutiva de Máquinas, basado en datos recopilados de estudiantes de conservatorio mediante la aplicación Sharpmony. La idea era analizar las acciones de los estudiantes, recopilando información útil que posteriormente pudiera aplicarse en la búsqueda de soluciones dentro del contexto de los AE, particularmente en una etapa de búsqueda local que podría incorporarse dentro del operador de mutación. En [9], esta idea fue descrita por primera vez, aunque solo se probó utilizando modelos sintéticos. A continuación, describimos cómo estas ideas fueron puestas en la práctica en conservatorios de música.

## 3. Metodología: Ayudando a profesores y estudiantes mientras ellos nos ayudan

Como se describió anteriormente, trabajamos en un ciclo de vida iterativo donde la mejora del proceso de enseñanza/aprendizaje, tal como lo aplican profesores y estudiantes, está entrelazada y es inseparable de la mejora del AE dedicado a la armonización a cuatro voces. Este proceso se ha desarrollado durante los últimos cinco años: más de 3.700 usuarios se han registrado y contribuido, y se han corregido más de 17.000 ejercicios. Primero analizaremos el problema y luego las dos caras de nuestra moneda.

### 3.1. El problema

El problema de la armonización a cuatro voces representa el primer acercamiento del estudiante a la composición musical. Si lo consideramos desde el punto de vista de los AG, el proceso comienza con una melodía predeterminada compuesta por una serie de compases (en el ejemplo de la Figura 2, se utilizan ocho compases), y el AE debe encontrar una partitura armonizada para SATB.

El proceso de armonización consta de dos etapas evolutivas secuenciales. En la primera etapa, el primer algoritmo determina los grados de la escala que se asignarán a cada nota de la melodía (por ejemplo, asignando II, V o VII a una nota re en la tonalidad de do mayor). En la segunda etapa, se determina la distribución de las notas del acorde entre las demás voces, asegurando que se utilice el grado previamente asignado.

Durante la primera fase, los individuos en el AG se representan mediante secuencias de grados de escala (I, II, ..., VII), asignando un grado a cada nota de la melodía. El objetivo es generar una secuencia de grados de acordes que emule el estilo compositivo comúnmente practicado por los estudiantes. El algoritmo asume que la melodía pertenece a una única tonalidad, como do mayor, sin modulaciones. Aunque las modulaciones (por ejemplo, de do mayor a la menor o sol mayor) son posibles, este estudio se centra exclusivamente en soluciones dentro de una única tonalidad, específicamente do mayor.

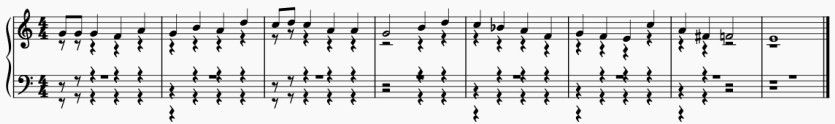

Figura 2: Melodía proporcionada al algoritmo para realizar la armonización.

Si bien en la armonización se pueden emplear muchos tipos diferentes de acordes, este enfoque considera únicamente tríadas diatónicas, acordes de séptima, acordes de sexta aumentada (Italiano, Francés, Alemán y Napolitano) y dominantes secundarias, siguiendo las reglas de la armonía.

En la segunda etapa, un AG independiente evoluciona la partitura musical completa. Aquí, cada cromosoma está compuesto por todas las notas asignadas a las cuatro voces de la partitura. Las notas se seleccionan para alinearse con los grados de acordes determinados en la primera etapa. Por ejemplo, para formar una tríada de grado I en do mayor, el algoritmo asigna do, mi y sol. La Figura 3 ilustra la estructura del cromosoma en ambos AG y sus interrelaciones.

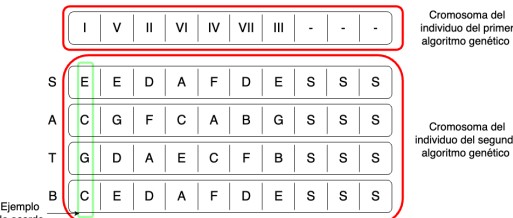

Figura 3: En la parte superior se representa el cromosoma de un individuo del primer AG, considerando la tonalidad de do mayor. En la parte inferior se muestra el cromosoma del individuo del segundo AG. Los nombres de las notas se dan en formato anglosajón. Se adopta la letra S para el silencio.

## 3.2. Función fitness

Aunque en la primera etapa una función fitness se encarga de determinar una progresión de acordes adecuada, aquí nos enfocamos en la segunda etapa, la principal, que se encarga de analizar las notas asignadas a los acordes.

Cuando las notas del acorde se distribuyen entre las demás voces según los grados previamente evolucionados, es necesario revisar toda la partitura utilizando la función fitness, que incorpora todas las reglas de armonía que comúnmente son aplicadas. Esta función evalúa hasta 50 tipos de errores diferentes, entre ellos:

- Quintas paralelas.
- Octavas paralelas.
- Entrecruzamiento de voces.
- ...
- (hasta 50 reglas, como se describió anteriormente)

Algunas de las reglas disponibles determinan si un conjunto de acordes es válido o no, como las dominantes secundarias, los acordes de séptima, los acordes napolitanos, los acordes de sexta, etc. Cuando se añaden nuevos tipos de acordes (como los acordes menores melódicos en 2023 o el acorde SUS4 en 2024; ver la Figura 4b, que muestra 290 informes enviados por los profesores a lo largo de los años para mejorar la función fitness), el espacio de búsqueda se expande. Debemos recordar que el enfoque iterativo desarrollado durante los últimos cinco años ha permitido agregar progresivamente estos nuevos tipos de acordes cuando los profesores los han solicitado. Sin embargo, esto tiene como consecuencia que el problema cada vez sea más complejo desde la perspectiva del AE, ya que el espacio de búsqueda se amplía y se debe aplicar un mayor número de reglas a cada par de acordes consecutivos, lo que a su vez incrementa el tiempo de cómputo.

### 3.2.1. Función fitness como ayuda para profesores y estudiantes

Sin embargo, si consideramos cómo funciona un AE, podemos notar que la función fitness es útil para evaluar cualquier ejercicio de armonización a cuatro voces, ya sea creado por un humano o por una máquina. Por ello, decidimos desarrollar una herramienta que permitiera a los estudiantes editar y revisar sus ejercicios de forma autónoma. Además, la herramienta permitió a los profesores seguir mejor el trabajo de sus estudiantes. En la siguiente sección, describimos el impacto de esta herramienta en el trabajo de los estudiantes.

### 3.2.2. Precálculo de la función fitness parcial

Tal como se describió anteriormente, cuanto más largo sea el ejercicio y mayor sea el número de reglas, más tiempo es necesario para calcular el valor fitness. En nuestros experimentos, con un

ejercicio de 29 acordes en 8 compases, analizamos el tiempo necesario para comprobar errores dentro de un par de acordes determinado: 22,5 segundos. Esto significa que analizar un solo ejercicio puede tomar alrededor de 10 minutos en el caso de ejercicios estándar realizados por los usuarios.

Para mitigar este desafío, intentamos anticipar el cálculo de los valores parciales de la función fitness correspondientes a posibles errores en acordes sucesivos y almacenarlos en una base de datos. Una vez disponible esta información, revisar el número de errores en un par de acordes sólo requiere el tiempo necesario para acceder a la base de datos: 0,001 milisegundos en lugar de los 22,5 segundos que se tardaría en aplicar todas las reglas. Aunque inicialmente planeamos analizar y almacenar todos los pares de acordes posibles, tras estudiar detenidamente la cantidad de combinaciones disponibles, notamos que el número de pares es tan grande que tomaría años calcularlos con la infraestructura disponible, como se describe en [9]: más de 8 millones de pares considerando una sola tonalidad, sin mencionar la complejidad añadida cuando hay modulaciones.

Sin embargo, recordemos que disponemos de una base de datos con los ejercicios de los estudiantes. Esta base de datos es crucial en nuestra estrategia: dado que el número de pares de acordes posibles excede las capacidades computacionales de las que disponemos, decidimos realizar un precómputo únicamente de aquellos pares que han sido utilizados en la base de datos, la cual contenía 13.000 ejercicios analizados en el momento del proceso de cómputo. Esto nos permitió restringir inicialmente nuestro espacio de búsqueda a 67.240 pares de acordes, en lugar de los 8.151.025 disponibles en una sola tonalidad. Para los lectores interesados, en [8, 9] detallamos cómo calcular el número de pares de acordes disponibles.

Específicamente, cada par de acordes fue almacenado en una base de datos con información sobre tonalidad, grado y número y tipo de errores en el par. Además, se registraron los tipos específicos de errores cometidos, lo que permite identificar con precisión los errores en cada par de acordes (por ejemplo, ausencia de la tercera en un acorde, cruce de voces, etc.).

Todos los experimentos se realizan en máquinas virtuales en un clúster Dell M1000e compuesto por 15 blades M600/M610. Los procesadores utilizados son modelos Intel Xeon, concretamente E5506, E5507, E5640, E5520 y X5670. Desde el inicio del proyecto en 2017, el clúster utilizado sigue siendo el mismo.

### 3.3. Añadiendo búsqueda local a la mutación dirigida

Se aplica la mutación dirigida, que restringe la mutación solo a los acordes consecutivos donde se han detectado errores. La idea es intentar corregir los errores manteniendo intactas las partes del cromosoma que son correctas.

Además, en lugar de seleccionar aleatoriamente un nuevo acorde para una posición determinada, se analiza los vecinos del acorde, es decir, los acordes cuyas notas están en posiciones cercanas en la partitura, para determinar cuál es la mejor opción considerando los errores que podrían surgir al introducir el nuevo acorde.

En la práctica, cada vez que se aplica la mutación, el algoritmo crea un conjunto de posiciones con errores y, a partir de estas, se elige un acorde para mutar. Se proponen dos modos de búsqueda local: (i) Se mantiene la tonalidad y el grado del acorde original, pero se busca en la base de datos una disposición diferente de las notas del acorde; (ii) Se conserva la tonalidad original, y el algoritmo puede elegir el acorde y grado más apropiados dentro de la misma tonalidad. En ambos casos, el objetivo principal es minimizar el número de errores.

Si se encuentra un par de acordes adecuado, se aplicará. De lo contrario, se intentará una búsqueda más extensa para encontrar un nuevo acorde. La búsqueda de un nuevo acorde, cuando actúa un operador de mutación, funciona de la siguiente manera: necesitamos un nuevo acorde en una posición determinada, pero este debe incluir la nota de la melodía ya existente, la cual no puede modificarse. Además, el nuevo acorde, al ser insertado, será analizado en su contexto y debe presentar el menor número de errores posible. Por lo tanto, en la búsqueda también consideramos el acorde previo e intentamos localizar en la base de datos pares de acordes donde dicho acorde previo sea el primero del par, seguido por otro acorde que contenga la nota dada en el soprano (melodía). Entre las opciones disponibles, seleccionaremos el par con el menor número de errores. Debemos recordar que esta información está disponible para cada par de acordes precalculado. Si no se encuentra un par adecuado, el sistema intentará generar y evaluar nuevos pares de acordes —un proceso mucho más

lento— para decidir qué acorde utilizar en el proceso de mutación. Además, la información de los nuevos pares generados se almacenará para futuras búsquedas.

## 4.   Resultados

En esta sección se analizan dos tipos de resultados diferentes. Por un lado, la mejora del algoritmo en la búsqueda de soluciones de armonización a cuatro voces. Pero, en segundo lugar, y aún más importante, como estas mejoras provocaron un cambio en la forma en que profesores y estudiantes desarrollaron una metodología asistida por IA, lo que, en última instancia, conllevó a la mejora del propio algoritmo.

### 4.1.   El impacto de Sharpmony en alumnos y profesores

Desde su lanzamiento en 2020, más de 3.700 usuarios se han registrado y se han editado y comprobado más de 17.000 ejercicios. La Figura 4a, muestra el número de ejercicios revisados por año, así como el crecimiento en la cantidad de usuarios. Podemos observar que cada año aumenta el número de ejercicios creados y comprobados, lo cual es positivo desde el punto de vista de la disponibilidad de datos para el aprendizaje. Además, la forma de la curva muestra indicios de cque la creación de ejercicios por año se está acelerando.

Debemos tener en cuenta que todos los ejercicios y la información sobre los errores detectados se almacenan en la base de datos del backend de la herramienta, lo que resultó decisivo para la mejora del AE, como describimos a continuación.

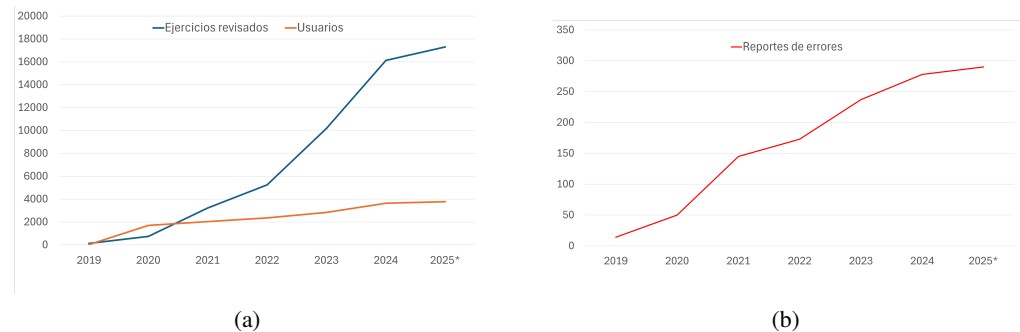

(a)                      (b)

Figura 4: (a) Número acumulado de ejercicios revisados y usuarios registrados a lo largo de los años. (b) Número acumulado de reportes enviados por los profesores para mejorar la herramienta a lo largo de los años.

Cuando preguntamos a los profesores cuál era el comportamiento de los estudiantes antes de utilizar la herramienta asistida por IA, específicamente sobre el número promedio de intentos que un estudiante realizaba en cada ejercicio propuesto, este valor era cercano a 1. Es decir, el estudiante intentaba resolver el ejercicio una vez y, solo en algunos casos, después de que el profesor revisara el ejercicio y le solicitara mejoras, intentaba una segunda vez. Por lo tanto, el número promedio de intentos por ejercicio era aproximadamente 1, lo que es similar a cualquier otra materia enseñada de manera tradicional: el profesor propone un ejercicio, el estudiante intenta resolverlo para la próxima clase y, en esa sesión, el profesor presenta la solución. Algunos estudiantes ni siquiera intentan resolverlo.

Sin embargo, si analizamos el número de ejercicios en la base de datos en el momento de escribir este artículo, observamos que se han creado 7.776 ejercicios individuales y se han revisado un total de 17.133 versiones de esos ejercicios en Sharpmony. Esto significa que, en promedio, se han realizado 2,20 versiones diferentes de cada ejercicio, lo que representa una mejora clara de más del 100 % en comparación con el valor anterior a la adopción de la herramienta asistida por IA.

En la Figura 5 se muestra un gráfico donde los ejercicios de los estudiantes se agrupan según la cantidad de veces que han intentado resolverlos. Además, distinguimos entre los estudiantes que pertenecen a instituciones que han adoptado Sharpmony (Figura 5a), utilizando todas las herramientas proporcionadas para profesores y alumnos, y los usuarios que emplean Sharpmony sin ninguna supervisión docente, como se muestra en la Figura 5b.

Lo primero que observamos es una diferencia interesante: más del 50 % de los ejercicios fueron intentados 2 o más veces por estudiantes supervisados, mientras que solo el 40 % en el caso de los no supervisados (ver los porcentajes asociados a cada barra). Calculamos el número promedio de veces que los ejercicios son intentados por cada grupo de estudiantes: 2,275 para los usuarios supervisados y 2,078 para los que trabajan de manera autónoma.

Curiosamente, la presencia de profesores aumenta en un 10 % el número de intentos para resolver los ejercicios. La principal conclusión es que las herramientas de armonización a cuatro voces asistidas por IA, en conjunto con la supervisión docente, permiten maximizar la práctica en este contexto específico de enseñanza musical, y la práctica es el camino hacia el aprendizaje.

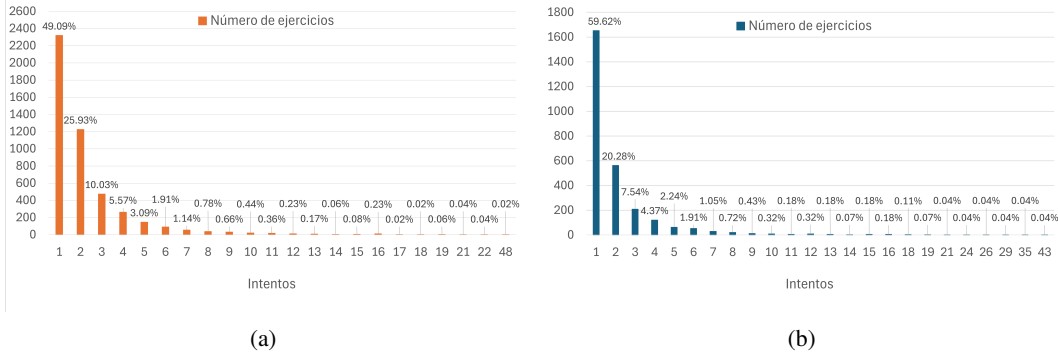

(a)             (b)

Figura 5: (a) Número de intentos para resolver un ejercicio dentro de conservatorios y universidades. (b) Número de intentos realizados por usuarios que trabajan solos.

Hemos consultado a los profesores sobre este comportamiento y nos han comentado un descubrimiento interesante: dado que el proceso de corrección puede aplicarse a los ejercicios tantas veces como se desee, los estudiantes desarrollan una estrategia sencilla: intentan mejorar progresivamente los ejercicios y eliminar los errores detectados, realizando tantos intentos como sean necesarios. Tal vez esta sea la razón por la cual algunos ejercicios han sido intentados más de 20 veces, algo que nunca había sucedido antes.

### 4.2. Usando los resultados de los estudiantes para mejorar la armonización a 4 voces evolutiva

Para aplicar la metodología propuesta y comparar los resultados actuales con los obtenidos previamente, se realizaron 35 ejecuciones con una población de 8 individuos. En la primera etapa, cuando se evoluciona la progresión de acordes, el AE se ejecutó durante 50 generaciones. En la segunda etapa, cuando las notas de los acordes se distribuyen entre las voces, se calcularon 10 generaciones nuevamente con una población de 8 individuos. Se emplearon un método de selección por ruleta y mutación dirigida en la segunda etapa. En ambas etapas se aplicó una probabilidad de cruce del 50 %. Inicialmente, la base de datos solo incluía los pares de acordes derivados del estudio del espacio de soluciones de los estudiantes.

Como se describió anteriormente, la disponibilidad de una base de datos con ejercicios de estudiantes permitió analizar e incluir información útil en la base de datos, lo que finalmente posibilitó la inclusión de búsqueda local dentro del AE. Se recopilaron un total de 67.240 pares de acordes a partir de los ejercicios disponibles. Junto con los pares de estudiantes ya almacenados, el algoritmo que armoniza la melodía puede generar nuevos pares cuando no se encuentra uno adecuado entre los pares registrados de los estudiantes. Sin embargo, se da prioridad a los pares provenientes de los estudiantes.

### 4.2.1. Espacio de soluciones actualizadas

Durante el proceso evolutivo, el AG concatena los acordes creando partituras musicales que contienen pares almacenados en la base de datos y, en algunos casos, combina los acordes, generando nuevos pares. En este caso, los nuevos pares se añaden a la base de datos junto con el número de errores que la función fitness ha calculado y una etiqueta específica que nos permite distinguirlos de aquellos provenientes de los ejercicios de los estudiantes. Estas etiquetas nos permiten analizar la base de datos resultante de los experimentos.

Al finalizar los experimentos, la base de datos consta de 107.774 pares de acordes, de los cuales 40.534 son nuevos, producidos por el AE, y 67.240 fueron recopilados inicialmente de los ejercicios de los estudiantes, representando el 37,61 % y el 62,39 %, respectivamente. Esta base de datos se empleará en futuros experimentos, por lo que prevemos que seguirá creciendo y, así, permitirá acelerar los experimentos en el futuro, dado que una mayor cantidad de pares ya estarán precomputados.

### 4.3. Ejecuciones

La Figura 6a ilustra la convergencia del algoritmo en tres configuraciones diferentes: (i) sin búsqueda local; (ii) con búsqueda local que busca identificar acordes apropiados dentro de la misma tonalidad y grado; y (iii) con búsqueda local que explora la misma tonalidad pero permite flexibilidad en el grado utilizado. Después de confirmar la efectividad de la búsqueda local, realizamos más experimentos utilizando solo esta configuración durante 50 generaciones. Los resultados, presentados en la Figura 6b, muestran una aptitud promedio de 1,8 con una desviación estándar de 1,7. Identificamos cuatro soluciones libres de errores. La Figura 7 muestra un ejemplo de una solución sin errores.

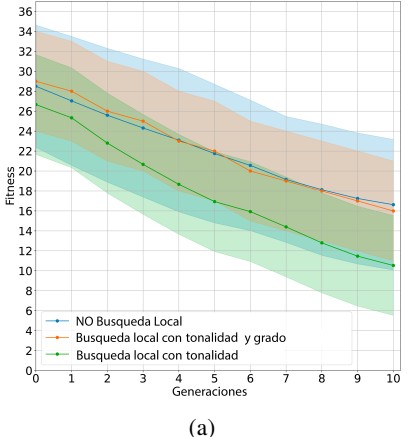
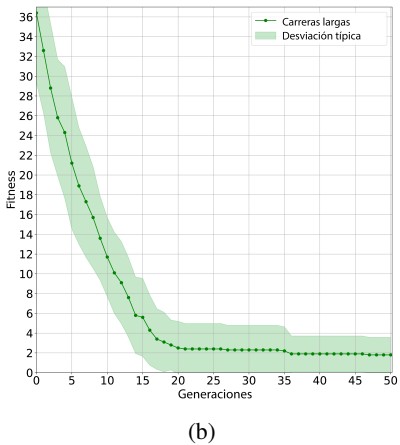

(a)                                                                                        (b)

Figura 6: (a) Convergencia de los valores promedio del mejor valor fitness entre las 3 versiones del algoritmo en 35 ejecuciones. (b) Convergencia del valor fitness en ejecuciones más largas.

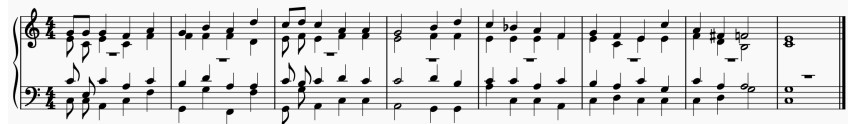

Figura 7: Armonización sin errores generada por el AE.

#### 4.3.1. Mutua mejora enseñanza/aprendizaje entre humanos y máquinas

Estamos siendo testigos de una mejora mutua tanto en el enfoque del AE para la armonización a cuatro voces como en la forma en que se enseña y aprende la materia al utilizar herramientas asistidas por IA. En primer lugar, los intentos de los estudiantes por resolver ejercicios han aumentado un 100 % desde la adopción de la herramienta asistida por IA, permitiendo que los ejercicios reflejen progresivamente lo que los profesores necesitan y solicitan a través de informes (como la incorporación del acorde SUS4 en 2024 o los acordes menores melódicos en 2023, con un total de 290 informes, como se muestra en la figura 4b). En segundo lugar, el creciente número de ejercicios realizados por los estudiantes y los informes enviados por los profesores para mejorar la herramienta han permitido que el AG explore de manera más efectiva espacios de búsqueda cada vez más amplios mediante la búsqueda local, logrando finalmente encontrar soluciones en un tiempo razonable.

## 5. Conclusiones

Este artículo describe una colaboración efectiva entre conservatorios de música e investigadores en IA que ha permitido desarrollar una herramienta capaz de resolver ejercicios de armonización a cuatro voces mediante enfoques evolutivos. Un componente clave del algoritmo, la función de evaluación, se incluyó en otra herramienta de enseñanza asistida por IA que permitió revisar automáticamente los ejercicios de armonización de los estudiantes. Esto ha permitido, ahorrar un tiempo valioso a los profesores, y ha conseguido que los alumnos practiquen más: el número de ejercicios realizados por los estudiantes aumentó en más de un 100 % en comparación con la metodología de enseñanza-aprendizaje tradicional. Más aún, cuando la herramienta se usó combinada con la enseñanza del profesor, se aumentó un 10 % adicional el número de ejercicios realizados, frente a aquellos que utilizaron la herramienta de forma independiente.

En segundo lugar, la colaboración con profesores e instituciones desde 2020 ha permitido ampliar el número de reglas y técnicas que el AE aplica durante la búsqueda de soluciones (290 mejoras a lo largo de los años). Además, logramos recopilar ejercicios de estudiantes, de los cuales se analizaron 13.000, almacenando pares de acordes útiles según el número de errores que presentaban. Esta información se utilizó en un enfoque de mutación dirigida + búsqueda local, que finalmente ha logrado, por primera vez, proporcionar soluciones libres de errores a ejercicios de armonización a cuatro voces en tiempos de ejecución razonables, demostrando así los beneficios mutuos enseñanza-aprendizaje humano-máquina que surgen en la educación asistida por IA.

## Agradecimientos

Agradecemos el apoyo del Ministerio de Ciencia e Innovacíon Español bajo los proyectos PID2020-115570GB-C21 y PID2023-147409NB-C22 financiados por MCIN/AEI/10.13039/501100011033. Junta de Extremadura en el marco del proyecto GR15068.

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
