# OpenReview forum: "Beneficios mutuos de la enseñanza-aprendizaje máquina-humano"
_MAEB/2025/Congreso — MAEB 2025_

### Official Review · Reviewer_k7ks · 2025-03-12
**Herramienta online con algoritmo evolutivo mejora eficiencia en ejercicios de armonía a 4 voces, aumentando intentos y retroalimentación rápida para estudiantes.**

**Rating:** 5
**Confidence:** 4

**Review:**

El artículo muestra una herramienta online basada en un algoritmo evolutivo para resolver ejercicios de armonía a 4 voces. Algunas de las ventajas del algoritmo, es la velocidad a la hora de resolver el ejercicio. Esto permite a los estudiantes de conversatorios poder tener retroalimentación más rápido y minimizar el trabajo de los profesores. Los resultados muestran como en vez de 1 intento por usuarios se realizan mínimo 2 intentos en promedio (2.20) y como algunos ejercicios se realizan de forma reiterada para tratar de mejorar.

Desde mi punto de vista, creo que es un problema interesante, el artículo está bien redactado y teniendo en cuenta algunos cambios menores debería ser aceptado para la presentación.

# Comentarios menores

- En el resumen se usa IA en la primera frase sin haber mostrado Inteligencia Artificial (IA).
- La línea 40 muestra "plantea una cuestión interesante", sería más adecuado hablar de hipótesis del trabajo.
- Línea 52, cuando se hablan de capacidades no entiendo si es en términos de tiempo. Parece hacer referencia a este motivo ya que luego habla del espacio de búsqueda (se deberá definir qué es el espacio de búsqueda).
- Línea 91 cambiaría eliminaría la palabra "fitness" ya que se habla de ella más abajo.
- Todas las figuras están en inglés, deberían ser "Figura", no "Figure".
- Descripción de la figura 7, se usa EA (entiendo que del inglés), debería ser AE. Adicionalmente, en la línea 351 se usa AE sin haberlo definido como "Algoritmo Evolutivo". Puedes hacerlo en la línea 97.
- Línea/s 325/336 creo que te refieres a 107.774 (no un valor decimal 107,774), lo mismo con 40.534 y 67.240.
- Figura 6.a: Busqueda -> Búsqueda; Local-> local; tontalidad -> tonalidad.
- Línea 389: References -> Referencias.

# Preguntas

- Desde 2017 han pasado 8 años y han mejorado los procesadores. ¿Qué características tenían los diferentes ordenadores en los que se ejecutaron los algoritmos?, si queremos comparar el tiempo, es importante conocer el entorno.
- Sección 3.2.2. - Entiendo que se usan 50 reglas, ¿qué tamaño tiene el ejercicio más largo? ¿cuánto tarda según el tamaño de los ejercicios? ¿Qué tamaño de ejercicios tenían los algoritmos previos?
- Sección 3.3. - Solo por confirmar, ¿la búsqueda local utiliza intercambios siempre y cuando sea adecuado/factible?
- La herramienta gamifica de cierta forma a los usuarios, haciendo que realicen más ejercicios o más intentos al mismo ejercicio. ¿Habéis estudiado si mejoran los conocimientos/aprendizaje? Sería interesante comparar a dos grupos de alumnos con el mismo profesor, un grupo con enseñanza tradicional y otro usando la herramienta. Comparar el número de intentos, tiempo dedicado, número de fallos y así poder validar cuánto aporta la herramienta. Hay algunas revistas (IEEE Transactions on Education (ToE); Expert Systems With Applications (ESWA) o IEEE Transactions on Learning Technologies (TLT)) que les interesaría el estudio.

---

### Official Review · Reviewer_cYYF · 2025-03-14
**Interesante transferencia de investigación a docencia**

**Rating:** 4
**Confidence:** 5

**Review:**

El artículo presenta una interesante aplicación de la investigación en el proceso de enseñanza-aprendizaje de lenguaje musical. El tema presentado es interesante y está bien escrito, pero existen algunos puntos de mejora que se podrían revisar, fundamentalmente de la narrativa, no de la investigación.

El abstract define el proceso y resultados, pero no menciona las técnicas concretas utilizadas.

La figura 3 no se ve bien. Se recomienda aumentar el tipo de letra del contenido de los cromosomas. También explicar la codificación del cromosoma del segundo individuo, ya que representa las notas en formato anglosajón (C -> do,…), y la S ¿para silencio?.

Los porcentajes de la figura 5 también son demasiado pequeños. Dado que se mencionan en el texto, se sugiere aumentar el tamaño de fuente.

En la sección 3.2.2., donde se indica el tiempo de evaluación, se agradecería precisar en qué tipo de computador, aunque se puede entender que se trataría de un ordenador personal o portátil de gama media.

La sección 4.2 describe la configuración de experimentos, pero no indica cuál es el objetivo. Es decir, ¿cómo contribuye el AG al incremento de información en la BD? ¿Es el mismo AG que se describe en la sección 3? Parece otro distinto, ya que la sección 3 busca generar armonías correctas. Se agradecería aclarar este punto.

Finalmente, el artículo excede la longitud prevista, que son 9 páginas sin contar referencias.

Erratas:

"Inteligencia Artificial" debería ir siempre en mayúscula.

Pg. 4: "corregido más de ejercicios"

Pg. 5: hay una referencia a la figura 5-b, pero parece que debería referenciar a la figura 4-b (también en la página 10?)

---

### Official Review · Reviewer_tPWT · 2025-03-18
**Contribución muy interesante, de un proyecto de 5 años de duración, en el que se aplican algoritmos evolutivos y colaboración IA-humano para la enseñanza de música**

**Rating:** 5
**Confidence:** 4

**Review:**

El trabajo es muy interesante. En él se describe un proyecto de 5 años de duración para la enseñanza-aprendizaje de música basada en algoritmos evolutivos. El proyecto se ha aplicado en entorno real, permitiendo obtener una gran cantidad de datos. Más importante aún, ha derivado en el diseño de herramientas que apoyan la enseñanza-aprendizaje de música, en concreto de la armonía a cuatro voces, basándose en un enfoque IA-humano que complementa a profesorado y estudiantado. Además, la metodología propuesta permite usar la información obtenida de los ejercicios de los estudiantes para mejorar a su vez el diseño de los dos algoritmos evolutivos considerados para resolver el problema.

El artículo está bien escrito y describe muy bien el proyecto. Presenta resultados previos y cómo se han ido aplicando al nuevo escenario. Los resultados son muy relevantes. En mi opinión, merece ser aceptado.

Algunos comentarios:

1) Hay una errata en la línea 153. No se indica el número de ejercicios considerados. Si no me equivoco, son 17000.

2) Sección 3.1: El lector entiende bien que se consideran dos algoritmos evolutivos para resolver el problema. Se intuye que la aplicación de ambos siempre es secuencial (podría ser también entrelazada). Estaría bien que se aclarara este aspecto.

3) La figura 6.a tiene dos erratas en el rótulo. Pone "tontalidad" en lugar de "tonalidad"

---

### Decision · Program_Chairs · 2025-03-20

Accept